# Highly Efficient CRISPR/Cas9 Mediated Gene Editing in *Ocimum basilicum* ‘FT Italiko’ to Induce Resistance to *Peronospora belbahrii*

**DOI:** 10.3390/plants12132395

**Published:** 2023-06-21

**Authors:** Marina Laura, Chiara Forti, Sara Barberini, Roberto Ciorba, Carlo Mascarello, Annalisa Giovannini, Luisa Pistelli, Ylenia Pieracci, Anna Paola Lanteri, Agostina Ronca, Andrea Minuto, Barbara Ruffoni, Teodoro Cardi, Marco Savona

**Affiliations:** 1CREA, Research Centre for Vegetable and Ornamental Crops, Corso degli Inglesi 508, 18038 Sanremo, Italy; marina.laura@crea.gov.it (M.L.); chiara.forti@ibba.cnr.it (C.F.); sara.barberini@ipsp.cnr.it (S.B.); roberto.ciorba@crea.gov.it (R.C.); carlo.mascarello@crea.gov.it (C.M.); annalisa.giovannini@crea.gov.it (A.G.); barbara.ruffoni@crea.gov.it (B.R.); teodoro.cardi@ibbr.cnr.it (T.C.); 2CNR-IBBA, Institute of Agricultural Biology and Biotechnology, Via Bassini 12, 20133 Milano, Italy; 3CNR-IPSP, Institute for Sustainable Plant Protection, Via Madonna del Piano 10, 50019 Sesto Fiorentino, Italy; 4CREA, Research Centre for Olive, Fruit and Citrus Crops, Via di Fioranello 52, 00134 Rome, Italy; 5Department of Pharmacy, University of Pisa, Via Bonanno 33, 56126 Pisa, Italy; luisapistelli53@gmail.com (L.P.); yleniapieracci@gmail.com (Y.P.); 6CeRSAA, Center for Agricultural Experimentation and Assistance, Regione Rollo 98, 17031 Albenga, Italy; anna.lanteri@rivlig.camcom.it (A.P.L.); agostina.ronca@rivlig.camcom.it (A.R.); andrea.minuto@rivlig.camcom.it (A.M.); 7CNR-IBBR, Institute of Biosciences and Bioresources, 80055 Portici, Italy

**Keywords:** sweet basil, CRISPR/Cas9, biotic stress, resistance, *ObDMR6*, élite cultivar, Genoese Pesto

## Abstract

*Ocimum basilicum* (sweet basil) is an economically important aromatic herb; in Italy, approximately 1000 ha of “Genovese-type” basil are grown annually in greenhouses and open fields and are subjected to Downy Mildew (DM) disease, caused by *Peronospora belbahrii*, leading to huge crop losses. Mutation of the Susceptibility (S) gene DMR6 (Downy Mildew Resistant 6) has been proven to confer a broad-spectrum resistance to DM. In this work, an effective Genome Editing (GE) approach mediated by CRISPR/Cas9 in *O. basilicum* ‘Italiko’, the élite cultivar used to produce “Pesto Genovese D.O.P”, was developed. A highly efficient genetic transformation method mediated by *A. tumefaciens* has been optimized from cotyledonary nodes, obtaining 82.2% of regenerated shoots, 84.6% of which resulted in Cas9+ plants. Eleven T0 lines presented different type of mutations in *ObDMR6;* 60% were indel frameshift mutations with knock-out of *ObDMR6* of ‘FT Italiko’. Analysis of six T1 transgene-free seedlings revealed that the mutations of T0 plants were inherited and segregated. Based on infection trials conducted on T0 plants, clone 22B showed a very low percentage of disease incidence after 14 days post infection. The aromatic profile of all in vitro edited plants was also reported; all of them showed oxygenated monoterpenes as the major fraction.

## 1. Introduction

*Ocimum basilicum*, known as sweet basil, is one of the major evergreen multipurpose aromatic herbs. It belongs to the Lamiaceae family and is considered allotetraploid (2*n* = 4*x* = 48) [1,2]. It is an economically important crop due to its chemical properties used in pharmaceutical, cosmetic, and culinary fields [3,4]. Leaves were extensively studied for its secondary metabolite content, which includes polyphenols such as flavonoids and anthocyanins [5], involved in important biological activities as antioxidant, anticancer, antimicrobial, anti-inflammatory, and insecticide. Furthermore, basil is appreciated for culinary flavoring and aromatherapy [6]. 

It is cultivated on commercial scale mainly in the Mediterranean area [7], and in Italy approximately 1000 ha of sweet “Genovese-type” basil are grown annually in greenhouses and open fields, in particular in the Riviera Ligure area (Liguria, Northern Italy) in order to produce the typical Italian Pesto sauce [8,9] recognized as a D.O.P. (Protected Denomination of Origin) product. One of the most used sweet basil varieties to produce Genoese Pesto sauce is ‘FT Italiko’. Conventionally propagated by seed [10], it is well-known and appreciated for its characteristic aroma used to flavor many dishes (https://www.topseed.info/, accessed on 4 April 2023). 

A severe problem affecting the massive cultivation and propagation of sweet basil is represented by plant susceptibility to *Peronospora belbahrii*, an obligate biotrophic oomycete, which causes basil downy mildew (DM) leading to a significant yield loss of the global production of the crop. In the last twenty years, infection of *Peronospora* heavily damaged basil cultivations causing crop losses close to 100% [4,11]. Up to now, many efforts have been taken to manage DM, most of which involves the use of chemical agents, with severe drawbacks in terms of human health and environment protection [12].

Thanks to traditional breeding, it is possible to transfer pathogen resistance to susceptible sweet basil cultivars and seed companies are focusing their attention to the development and commercialization of new resistant varieties [13]. Different *Ocimum* species display resistance and/or tolerance to DM; however, they differ greatly from sweet basil cultivars in terms of ploidy, phenotype, and aroma [12]. Furthermore, breeding approaches are often too long and may lead to a sexual incompatibility, sterility, and presence of undesirable traits in the hybrids [12,14,15,16]. 

Currently, with the development of Genome Editing technologies, it is possible to turn off Susceptibility (S) genes that make plants vulnerable to pathogens, ensuring resistance. The most widely applied system for plant genome editing is CRISPR/Cas9 (Clustered Regularly Interspaced Short Palindromic Repeats/CRISPR-associated protein 9), due to its easiness in use and effectiveness in targeting many plant species [17,18]. Most important, the development of homozygous or complete knockout mutants in the T1 lines has been reported in both diploid and polyploid species [19,20,21]. 

*DMR* (*Downy Mildew Resistant*) genes are involved in the host–pathogen interaction, and it is reported that their inactivation could improve plant resistance [22]. The DMR6 gene was discovered and characterized in *Arabidopsis thaliana* [23,24], and its mutation has been shown to confer resistance to oomycetes [25]. The role of *DMR6* was also investigated in *Solanum tuberosum* [26], *Vitis* spp. [27], and *Castanea sativa* [28]. DMR6 encodes a 2-oxoglutarate (2OG)-Fe(II) oxygenase, belonging to the Flavone synthase I class (FNSI), and is involved in salicylic acid (SA) catabolism that should be maintained in homeostasis conditions [29,30].

*DMR6* editing approaches were reported in *Solanum lycopersicum* [31], *Hordeum vulgare* [32], *Musa* spp. [33], *Solanum tuberosum* [34], *Malus domestica* [35], *Citrus* spp. [36], *Vitis vinifera* [37], and *Ocimum basilicum* ‘Genoveser’ [38], demonstrating its role in the resistance to downy mildew disease. 

The aim of this research was to perform *DMR6* editing approaches in the élite cultivar ‘FT Italiko’ of *O. basilicum* to obtain *P. belbharii* resistant cultivars without altering specific organoleptic characteristics, while reducing the use of pesticides.

## 2. Results

### 2.1. Flow Cytometry

According to flow cytometry (FCM) analysis, ploidy of ‘FT Italiko’ plants was compared with the position of the reference DNA peak (*O. tenuiflorum* = *O. sanctum*; 2n = 32) that is on channel 200 on *x*-axis. The peak of ‘FT Italiko’ was located in a range of 400 channel assuming the possible tetraploid content (2n = 48) (Appendix A). This was also confirmed by the mean of fluorescence ratio between sample and reference calculated (approx. 1.9). 

### 2.2. Identification of ObDMR6 in ‘FT Italiko’ and Protein Alignment

The sequence *ObDMR6* in ‘FT Italiko’, obtained by BLASTN analysis, reached a max score alignment (371) and with E value of 4 × e^−99^ and showed sequence homology with *AtDMR6*. The translation of transcript was confirmed with BLASTX search in non-redundant protein database, showing 88.10% of identity with 2-oxoglutarate and oxygenase superfamily protein of *Perilla frutescens* var. *frutescens* (GenBank accession: KAH6789542.1), 87.50% of identity with downy mildew resistance-like protein 6 v3 and v1 of *O. basilicum* cv ‘Genoveser’ (GenBank accession: QWT44769.1 and QWT44767.1, respectively). 

Whole genomic sequence of 1260 bp of *ObDMR6* in ‘FT Italiko’ was isolated and sequenced. The alignment with *AtDMR6* coding sequence suggested the presence and the position of four exons and three introns (Figure 1).

The resulting *ObDMR6* cds was predicted to have an ORF of 1008 bp encoding 336 amino acids. The *ObDMR6* ‘Ft Italiko’ sequence was deposited on NCBI GenBank with the accession number MT319764 and released on 30 September 2021.

The CoGeBlast analysis has located the blast hits in six chromosome scaffolds (262, 7503, 4335, 8208, 235, 9553), discriminating almost six versions of *ObDMR6*, as described for sweet basil cv ‘Genoveser’ by [38]. These sequences show different SNPs against *ObDMR6* ‘FT Italiko’ and among them, only scaffold 7503 and scaffold 262 presented high percentage of identities with query sequence, respectively, of 95% and 93%. 

‘FT Italiko’ *ObDMR6* amino acid sequence (QQL14521.1) shared high similarity with *ObDMR6* (v1, v3, v5) and *AtDMR6* with a percent identity of 87.50% and 66.47%, respectively, calculated using BLASTP against *nr* protein database. The alignment of six *ObDMR6* amino acid variants [38] and *ObDMR6* ‘FT Italiko’ showed 24 amino acid substitutions among the protein sequences (Appendix A). The amino acid alignment with AtDMR6 was shown in Appendix A.

*ObDMR6* shows domains belonging to PLN02639 superfamily (oxidoreductase, 2OG-Fe (II) oxygenase family protein) that was detected using NCBI Conserved Domain Search [39] (Appendix A).

### 2.3. Editing in Hairy Roots

To evaluate the efficiency of the CRISPR/Cas9 system in basil, genetic transformation experiments mediated by *Agrobacterium rhizogenes* were performed. A high induction of Hairy Roots (HR) was obtained from all explants tested (cotyledons, leaves, in vitro stems) (Appendix A). Twenty-nine out of thirty HR were found to be positive in the PCR analysis for the presence of Cas9, confirming a high rate of co-transformation equal to 96% (introduction of the T-DNA genes of the Ri plasmid and the pDirect 22C plasmid).

Sequencing of the 396 bp fragment comprising sgRNAs target site (Figure 2) of two Cas9^+^HR lines, allowed us to identify frameshift mutations caused by the deletion of a nucleotide (C), in position 5 nt upstream of the PAM (Figure 3).

### 2.4. Production of T0 Edited Plants by Agrobacterium Tumefaciens-Mediated Transformation

After four weeks of culture, it was possible to detect shoots that directly emerged from 150 CNs, without any visible callus formation (Figure 4A,B). Eighty-two percent of regeneration efficiency, calculated as the ratio between number of regenerated CNs on total number of CNs, was recorded on kanamycin containing medium, with a mean of 2.6 of regenerated shoots for each explant. CNs, excided from 4-week-old plantlets generated transgenic plants expressing sgRNA(s) and Cas9. 

Different clones were obtained by shoot multiplication (Figure 4C). After 30 days, all singularized plants were able to develop an efficient root system (Figure 4D). Interestingly, some in vitro clones bloomed (Figure 4E). All clones transferred in insect-proof greenhouse were acclimatized and three T0 lines (21D, 22B and 45B) flowered and produced seeds (Figure 4F). Twenty-six T0 plants were tested for Cas9 presence by PCR, and 84.6% resulted to be positive for the integration of the transgene.

To identify the mutations in 21 *ObDMR6* transgenic plants, a 396 bp fragment comprising both sgRNA target site (Figure 2) was amplified by PCR and subsequently Sanger sequenced for each samples analyzed. 

The sequence alignment of 396 bp fragments among fourteen WT individual clones of *ObDMR6* showed four different variants (Appendix A). The SNP (A/T) at 11 bases upstream the PAM leads to amino acid change in the sequence: tyrosine (Y)/phenylalanine (F). 

Different type of mutations in *ObDMR6* were identified in 86% of T0 lines transformed with pDirect22c_sg442-462 (Figure 4). 

Seven different mutation patterns were observed in the T0 plants. Seven out of fifteen T0 mutants had nucleotide deletions around the target site, while two mutants show insertions of one and two bases within target site. 

More than sixty percent (9 out of 15) of the T0 mutants were confirmed to have frameshift mutations that cause the knock-out of *ObDMR6* gene and the premature termination of gene product. Five out of seventeen of the mutant plants show one or more base substitutions leading to amino acid sequence changes (Appendix A).

### 2.5. Cas9-Free ObDMR6 Mutants in T1 Generation

Six T1 transgene-free seedlings (derived from 21D and 22B lines), screened by PCR, having eliminated the Cas9 gene by segregation, were analyzed for the *ObDMR6* mutation derived from editing and compared to WT, by direct Sanger sequencing of the 396 bp fragment (almost three fragments for each plant were analyzed). The chromatograms generated for each plant were compared to WT and among T0 and T1 of the same line. Plants showing chromatograms with overlapping spikes (two nucleotides simultaneously) were considered heterozygous.

The chromatograms obtained from each T1 plants reveal that the mutations of T0 plants were inherited and segregated. Plants of line 22B had the deletion of a nucleotide (C), in position 5 nt upstream of the PAM, leading to the creation of a premature termination (stop) codon. For the T1 plants analyzed, the mutation appears to be homozygous, although the chromatogram of plant T1_22B_6 also shows a heterozygous single nucleotide polymorphism (SNP) that is anyway outside of the editing region: two peaks of equal heights but different colors (A/T) (as highlighted with ** in Figure 5). Plants of line 21D had the substitution of two nucleotides CG in place of TT, in position 7 nt upstream from the PAM, resulting in amino acid change in the sequence: alanine (A) instead of valine (V). Plant T1_21D_1 inherited the same mutation of T0_21D and displayed further nucleotide changes upstream of the PAM (CG in place of TG). Furthermore, it showed the same heterozygous SNP of plant T1_22B_6 (also denoted by **). 

### 2.6. Morphological Phenotyping of T0 and T1 Plants

The morphological traits observed in T0 plants are reported in Table 1 and Appendix A. For each clone and WT were evaluated almost three plants. The clones 21D, 47C, and 45B showed strongly decrease in plant height, internode number and leaf length. The shorter internode length and curling up margin of some leaves characterized 21D and 47C clones. The clones 17A2 and 32A differed from WT ones with respect to their height. Their leaves showed a normal size, but few assuming a shape with curling up margin. Plants of 22B clone are similar to WT plants, except for the morphology of few leaves. Flowering was normal in 22B, 21D, and 45B, but absent in the other three clones.

In Table 2, the main morphological traits of T1 plants are reported. The clone 21D does not show morphological traits different from WT plants. The 22B clone differed statistically from WT for internode length (Appendix A).

### 2.7. P. belbahrii Infection Trial

As reported in Table 3 and Table 4, the values of Disease Incidence and Disease Severity increased in percentage from 7 to 14 dpi for all the clones, reaching the maximum value (100%) for WT, 17A, 47C, and 21D. Even none of the clones showed infection symptoms at the end of the trial, clones 47B and 22B displayed a medium level of tolerance to the infection. Particularly, at 7 dpi, clone 22B did not show any infection symptom (0% of DI) and very low percentage of incidence and severity were detected at 10 and 14 dpi, in respect to the others clones and WT, confirming that this clone could be very promising. Visual symptoms on leaves of the clones 47B and 22C, showing the absence of spots and or chlorosis on the underside of the leaves and compared to the WT are reported in Figure 6.

### 2.8. Volatile Profile Analysis 

The complete chemical compositions of the headspaces (HSs) of the analyses *O. basilicum* clones are reported in Table 5. A total of 36 compounds were identified, accounting for 99.6–100.0% of the whole compositions. The class of terpenes was the most represented, followed by phenylpropanoids. Among the former class, monoterpenes were detected in good amounts in both their hydrocarbons and oxygenated forms, even though the latter was sharply predominant, accounting for 33.8–64.5%. The control sample was characterized by the highest amount, followed by the pairs 21D (47.4%) and 47C (47.1%) and 17A2 (39.4%) and 45B (36.3%), each with similar contents of oxygenated monoterpenes. Conversely, the sample 22B (33.8%) showed the lowest relative abundance of these secondary metabolites. 1,8-Cineole was the most important compound detected in all the samples, whose content ranged from 29.9 to 44.1%. Linalool, instead, was found in percentages above 15% in the control HS, while all the other samples showed narrowed contents, always below 5%. 

Monoterpene hydrocarbons, mainly represented by (*E*)-β-ocimene (3.2–11.2%) e β-pinene (2.2–4.9%), were also found in not negligible amounts, as they ranged from 16.6 to 24.6% of the HSs. The highest content of this class was detected in the 21D HS, while the significantly lowest in those of 45B and CTR. 

Among terpenes, also sesquiterpene hydrocarbons represented an important chemical class, whose content was comprised between 8.1 and 21.4%. These compounds were minimum in the control sample, while their content was greatest in 45B. All the other samples, instead, did not show significant differences. Among sesquiterpene hydrocarbons, *trans*-α-bergamotene was certainly the most relevant one.

Finally, the class of phenylpropanoids was detected in percentages from 10.7 in CTR up to 29.4% in 22B. This class was represented only by eugenol and its methylated derivative. The former compound was higher in all the analyzed clones, with the only exception of 45B, in which, instead methyl eugenol prevailed. 

The complete chemical composition of the HSs was subjected to multivariate statistical analysis with the Hierarchical Cluster Analysis (HCA) and the Principal Component Analysis (PCA) methods whose dendrogram and plots are reported in Figure 7 and Figure 8, respectively.

The dendrogram of the HCA was characterized by two main clusters, the red and the green ones. The former comprised the samples CTR, 21D, and 47C, of which the last two showed a great similarity, gathering in the same subgroup. The same was found for 17A2 and 22B, which along with 45B, formed the green cluster. 

A similar partition was also evidenced by the score plot of the PCA (Figure 8a), in which the samples belonging to the green group of the dendrogram were plotted in the left quadrants (PC1 < 0), while those of the red cluster in the right ones (PC1 > 0). Regarding the former group, the positioning of the samples 22B and 17A2 was, perhaps, determined by the greatest content of eugenol, while that of 45B by methyl eugenol. Conversely, the samples of the red group were probably plotted in the right quadrants on behalf of the 1,8-cineole vector. However, the positioning of the control sample in the rightmost area of the upper quadrant (PC2 > 0) was attributable to the linalool vector, whose content was significantly greatest in its HS.

Both statistical analyses evidenced a higher similarity of the control sample to the clones 47C and 21D, in terms of spontaneous volatile emission, as they are all characterized by both a greater content of oxygenated monoterpenes and a lower amount of phenylpropanoids. 

## 3. Discussion

Climate change is causing rapid changes in the intensity and spread of biotic stresses, such as old and new parasites [40]. It is, therefore, essential to have adequate technologies to develop new plant genotypes endowed with durable and broad-spectrum resistance. Traditional breeding does not always succeed in producing new genotypes capable of coping with rapid changes. Gene-editing technology makes possible to modify target DNA in a specific site, in a precise and fast way, inducing specific genetic variations, but maintaining the genetic identity [41] of elite cultivars. The inactivation of susceptibility genes required host–pathogen recognition or supporting infection process at different step and has allowed us to obtain plants resistant to different pathogens, in several economically important crops. Furthermore, the CRISPR/Cas9 technology makes it possible to generate transgene-free gene-edited plants resistance, which are better accepted by public opinion and do not have to fall back into European GMO legislation [42].

In this work, an effective Genome Editing approach mediated by CRISPR/Cas9 in *O. basilicum* ‘Italiko’, the elite cultivar used to produce “Pesto Genovese D.O.P”, was developed. 

The regeneration of whole plant from modified cells and tissues, through in vitro culture techniques, represents the main bottleneck for the application of Genome Editing in different plant species. A high rate of regenerated shoots is a necessary requirement for efficient genetic transformation. High percentage of sprouts regenerated by the CNs in *O. basilicum* cv ‘Italiko’ was recorded, confirming that cotyledonary node-based direct regeneration method is effective and suitable for genetic transformation mediated by *A. tumefaciens* [43]. Furthermore, direct shoot organogenesis, without callus proliferation, shortens the time to obtain regenerated plants and decreases the risk of somaclonal variation [44], facilitating the selection of transgenic plants.

A successful transformation protocol mediated by *A. rhizogenes,* carrying pDirect 22C binary vector and Ri plasmid, allowed us to obtain HRs with a high rate of co-transformation. Several studies reported the effective application of *A. rhizogenes* system in genome editing [45,46]. HRs, thanks to their prolific and quick growth rate and being easy to maintain for long time, represent a robust tool to test the efficiency of CRISPR/Cas9 system. Furthermore, the recalcitrant-to-regeneration plants could display after genome editing a good shoot regeneration potential [47].

Several studies conducted in different species of plants and crops (tomato, potato, barley, banana, citrus, apple, grapevine, basil) have shown that the mutation of S gene *DMR6*, discovered and characterized in *A. thaliana* [24], conferred resistance to oomycete and other pathogens. Hasley and colleagues [38], targeting *ObDMR6* in sweet basil ‘Genoveser’ by CRISPR/Cas9, found enhanced DM resistance with reduced production of sporangia and pathogen accumulation.

The sequence of *ObDMR6* from ‘FT Italiko’ was deposited on NCBI GenBank at the same time with the release of sequences of sweet basil *DMR6* cv Genoveser, (Enza Zaden) [38]. The comparison with the latter and with the *O. basilicum* genome sequences recently released [48] indicated the presence of six variable copies of the *DMR6* gene in basil, being the ‘FT Italiko’ sequence used in the present study closer to copies present in genomic scaffolds 7503 and 262. *O. basilicum* presents high interspecific variability even among cvs, in terms of morphological and chemical traits (aromatic profile) and susceptibility or resistance gene against pathogens [1,49]. 

The two overlapping sgRNA target sequences were designed on the second exon of the gene and were found to be conserved in all six chromosome scaffolds identified by CoGeBlast tool. It is reported that the overlapping sgRNAs (that share some base pairs) create small nucleotide deletions in target site with higher efficiency [50]. The presence of a SNP (A/T) 11 bases upstream one of the two PAMs in a sample of 14 wild type ‘FT Italiko’ plants should not reduce Cas9 ability to bind and cleave its target [51]. Indeed, different type of mutations in *ObDMR6* were identified in 86% of tested T0 transgenic plants; the 60% are indel frameshift mutations with knock-out of *ObDMR6* ‘FT Italiko’ and essentially two types of deletions (−1 and −2) and insertions (+1 and +2). As reported in two recent manuscripts [38,52], although basil is tetraploid, it has been possible obtain homozygous mutant plants free of the transgene at T1 generation also in *O. basilicum* cv ‘Italiko’. Chromatograms generated for each plant of two T1 lines (21D and 22B) clearly showed that mutations were inherited from their parents and were homozygous. Differences between the sequences of T0 plants and T1 progenies may be due to Cas9 activity between the time of tissue sampling of T0 plants and gene segregation following meiosis. Additionally, ref. [53] reported different mutation pattern between T1 plants and their own parent plants.

The evaluation of some morphological traits measured on acclimatized in primary transgenic plants (T0), showed differences among clones and WT, probably as a consequences of in vitro culture and/or *DMR6* gene editing. Only the 22B clone appears similar to WT, for plant and leaf traits. Few papers described the phenotype of *DMR6* mutants in different crops. In potato, an affection in growth was reported in phenotypes for *StDMR6-2* mutants, with significantly lower plant height and shorter internodes [34]; while in tomato [31], *SlDMR6-1* mutants displayed indistinguishable phenotype from WT plants under laboratory conditions. No phenotype description in sweet basil has been reported for *ObDMR6* mutants [38].

To the best of our knowledge, this is the first study reporting the chemical composition of the spontaneous volatile emission of in vitro plants of *O. basilicum* ‘FT Italiko’ and the first concerning the aroma of edited sweet basil plants. The aroma composition represented one of the most important aspects of *O. basilicum* cultivars used for the preparation of pesto sauce, as it is responsible for its organoleptic properties. 

The main chemical compounds detected in the control sample partially corroborate the findings of [54] who reported a prevalence of linalool, instead of 1,8-cineole in the volatile composition of *O. basilicum* cv. Genovese. In [55], oxygenated monoterpenes are the major secondary metabolites found in vitro plants, with abundance of linalool and 1,8 cineole depending on the different culture media used. According to our findings, gene editing was able to significantly influence the aroma profile of basil plants, creating two groups of samples: the former including the clones 21D and 47C, more similar to the control sample, but with a marked lower content of linalool on behalf of eugenol, and the second, including the clones 17A2, 22B, and 45B, in which the synthesis of methyl eugenol was strongly enhanced. Methyl eugenol is a common phenylpropanoid found in several plant species, whose release is seemingly related to the plant defense against pathogens and herbivore insects [56]; this information could be related to the fact that plants 45B and 22B are less susceptible to the *Peronospora* infection. 

In the experimental trial with *P. belbahrii* fresh spore solution, under permissive conditions, T0 22B and 21D plants have shown an enhanced DM resistance than WT, confirming the involvement of the *DMR6* gene in the interaction and recognition with its pathogen and that its inactivation improves resistance to DM. Quite significant differences are observed among Cas9+ plants in disease resistance, morphological traits and aromatic profile are probably due to intra-variety variability of commercial seeds as starting material for obtaining CNs and to independent origin of each regenerant plant from CNs following genetic transformation. Furthermore, other factors should be considered, as non-predictable integration of non-T-DNA portions of the vector into the plant genome [57] due incorrect recognition of borders by Agrobacterium and off-target effects on the genome that were predicted only in silico. In addition, multiple members of the DMR6-like gene family, recently identified in sweet basil [38,58], may contribute to showing varying degrees of silencing and partially restore the susceptibility to DM. In the future, the progenies derived from these promising plants will have to be tested in field trials to evaluate their potential in comparison to current and original cv and subsequently used in breeding programs to create new lines of elite cv sweet basil resistant to DM.

## 4. Materials and Methods

### 4.1. Flow Cytometry

Ploidy level of *O. basilicum* ‘FT Italiko’ was estimated using cytometry. Cell nuclei were isolated from upper young leaves of basil plant, together with reference plant (*O. tenuiflorum* = *O. sanctum*, 2C = 0.72 pg), by chopping leaves with a sharp razor blade in the Petri dish in 500 µL of nuclei extraction buffer (CyStain-UV precise P, Sysmex-Partec). To reduce the negative effect of cytosolic and phenolic compounds, 1% of PVP-40 (polyvinylpyrrolidone-40; Sigma PVP40) was added to the buffer [59]. Then, the crude suspension was filtered through a 30 µm CellTrics disposable nylon filter and was kept on ice for 10 min. For samples coloring, 1.6 mL of DAPI staining solution (4′,6-diamino-2-phenylindole) was added, according to the manufacturer’s instructions. The fluorescence intensity was measured by CyFlow Ploidy Analyser (Sysmex-Partec), equipped with a UV LED chip (365 nm) for DAPI excitation. The flow rate varied between 20 and 50 events per second, and about 6000 nuclei per sample were measured. Histograms were acquired using CyView software (Sysmex-Partec). Reference peak was gated manually, and coefficient of variation [60] and maximum peak height were recorded. The mean of fluorescence ratio between sample and reference was used to estimate the DNA ploidy levels. Analyses were performed twice for each sample (5 leaf samples of *O. basilicum* ‘FT Italiko’ and 5 leaves of *O. tenuiflorum* plants).

### 4.2. Regeneration Protocol Via Direct Organogenesis and Agrobacterium Tumefaciens Mediated Transformation 

Seeds of *Ocimum basilicum* L. ‘FT Italiko’ (La Semiorto Sementi^®^) were sterilized and germinated according to [61] and cotyledonary nodes (CNs) were used as starting material for plant transformation and direct shoot formation as suggested by [62] on regeneration medium (4.3 g/L MS “shoot multiplication medium”, 3% sucrose, 3.7 mg/L TDZ, 8 g/L agar, pH 5.8). 

*A. tumefaciens* was cultured in 5 mL of liquid TYNG medium (10 g/L tryptone, 5 g/L yeast extract, 5 g/L NaCl, and 0.5 g/L MgSO_4_·7H_2_O, pH 7.5); supplemented with rifampicin 50 mg/L, kanamycin 100 mg/L, and carbenicillin 100 mg/L; and incubated at 28 °C for 24 h at 120 r.p.m. The culture was incubated overnight under the same conditions and then centrifuged at 2000× *g* for 15 min at room temperature. Pellet was resuspended in MS liquid medium up to OD600 of 0.25–0.3; acetosyringone was added to a final concentration of 100 μM before plant inoculation. 

One-hundred fifty CNs were wounded, immersed for 30 min in suspension of *A. tumefaciens* cells, blotted dry and then placed on regeneration medium. After three days of co-cultivation, CNs were transferred to a regeneration medium supplemented by cefotaxime 100 mg/L and kanamycin 30 mg/L and sub-cultured every two weeks. 

Regenerated shoots developed from the CNs were excised and cultivated on a multiplication medium (4.3 g/L MS, 3% sucrose, 0.2 mg/L 6-Benzyladenine (BA), 8 g/L agar, pH 5.7) with addition of cefotaxime 100 mg/L and kanamycin 50 mg/L. Each shoot was considered as a single clone. All plant materials were maintained in vitro at 23 ± 1 °C with a 16 h photoperiod of light (PPFD 30 μE m^−2^ s^−1^). 

### 4.3. Agrobacterium Rhizogenes Mediated Transformation 

*A. rhizogenes* strain LB9402 was cultured in 5 mL Nutrient Broth (NB) medium supplemented with kanamycin 100 mg/L according to the protocol described above. 

Different explants (leaves, internodes, hypocotyls, petioles) were excised from seedlings, wounded, and immersed in bacterial suspension for 20 min at room temperature and then dried with sterile filter paper. The explants were placed on MS solid co-cultivation medium for 3 days at 24 °C in dark condition and subsequently placed on the same substrate with the addition of cefotaxime 100 mg/L and kanamycin 50 mg/L. After two weeks, 30 hairy roots (of 3 cm) were excised and kept separate as independent lines; they were sub-cultured every 15 days onto MS medium enriched with antibiotics.

### 4.4. Identification of ObDMR6 in ‘FT Italiko’

In order to identify *ObDMR6* in ‘FT Italiko’, *AtDMR6* coding sequence was used as a query (GenBank accession: NM_122361) against the Transcriptome Shotgun Assembly Sequence Database of *Ocimum tenuiflorum* [63]. The sequence with the highest alignment score and with 4 × e^−99^ value was checked in NCBI (https://www.ncbi.nlm.nih.gov/, accessed on 1 March 2021) non-redundant protein database (nr) using a BLASTX search, to confirm the translation. 

Primers, targeting region surrounding plant start codon (ATG) and the region comprising the stop codon (TAA) were designed and used to amplify *ObDMR6* of sweet basil ‘FT Italiko’ by PCR (DMR6_Fw: 5′-ATGGAAAATAAGGTGATTAG-3′; DMR6_Rv:5′-ATTCTTGAACAGTTCTAGGC-3′).

Genomic DNA was extracted from fresh ‘FT Italiko’ basil leaves, using Dneasy^®^ Plant Mini Kit (Qiagen, Hilden, Germany) according to manufacturer’s protocol and used as template to perform PCR with Q5 High-Fidelity DNA Polymerase (NEB, Ipswich, MA, USA) following these conditions: initial denaturation at 98 °C for 2 min; followed by 35 cycles at 98 °C for 30 s, 55 °C for 30 s and 72 °C for 30 s, with a final extension at 72 °C for 4 min. 

The PCR product (1260 bp) was cloned in pJET1.2/blunt vector (ThermoFisher^®^, Scientific Inc., Waltham, MA, U.S.) and subjected to Sanger sequencing. The genomic nucleotide sequence was aligned with *AtDMR6* cds to detect exons and introns position; the resulting *ObDMR6* cds was translated using Expasy tool (www.expasy.org, accessed on 15 April 2021) and the amino acid sequence was searched in nr protein database using BLASTP to assess its homology. The amino acid sequence alignment was also generated using CLUSTALW tool of Bioedit 7.2 software (https://bioedit.software.informer.com, first access on 15 April 2021).

With CoGeBlast, *ObDMR6* cds was compared against *O. basilicum* ‘Perrie’ genome (ID 59011) deposited in CoGe database (https://genomevolution.org/coge/, accessed on 10 January 2023).

### 4.5. Protein Alignment

The alignment of six ObDMR6 amino acid variants (QWT44767.1, QWT44768.1, QWT44769.1, QWT44770.1, QWT44771.1, QWT44772.1; (Hasley et al., 2021 [38]) and ObDMR6 homolog in *O. basilicum* ‘FT Italiko’ (QQL14521.1) was performed using ClustalW Multiple alignment tool of Bioedit program 7.2.5 [64]. 

### 4.6. Identification of CRISPR/Cas9 Targets and Vector Construction 

Two sgRNA target sequences were identified in the coding region by CRISPR Direct design tool (https://crispr.dbcls.jp/, accessed on 15 April 2021) [65] and selected for editing ‘FT Italiko’ *ObDMR6*: sgRNA442 (target sequence CAGGCTTCATTGCTACCCCT) and sgRNA462 (target sequence TGGAGAATTATGTTCCTGAA) directed to the second exon of the gene). The sgRNA target sequences are specific for scaffold 7503 but presents a SNP in position 10 of sgRNA462 target for the scaffold 262 (CoGe database).

pDirect_22C_sgRNA442-462 vector (35S:Csy4-P2A-AtCas9 + CmYLCV: gRNA-array) expressing a Csy4 array of two gRNAs targeting *ObDMR6* was constructed by direct assembly of gRNAs into the T-DNA vector pDirect_22C (Addgene plasmid 91135), according to protocol 3A of [66]; this vector includes *Arabidopsis thaliana* codon-optimized versions of Cas9 for use in dicots. 

The vector was sequenced in order to verify the correct insertion of the cassette, then it was introduced into *A. tumefaciens* strain AGL1 [67] for *O. basilicum* ‘FT Italiko’ transformation, according to the protocol described by [68]. pDirect22c_sg442-462 was also introduced in *A. rhizogenes* strain LB9402 to evaluate the efficiency of genome editing in the élite cv of *O. basilicum* hairy roots system. 

### 4.7. Screening of Transgenic Plants and HRs for Target ObDMR6 Mutations 

Genomic DNA was extracted from 50 mg of HRs and from 100 mg leaf tissue of newly regenerated basil plantlets (T0) and wild-type plants, using DNeasy^®^ Plant mini-Kit, (QIAGEN, Hilden, Germany). In order to detect transgene integration, PCR was performed using primers Cas9_Fw and Cas9_Rv (5′-GTGCAGACCTACAACCAGCT and 5′-CTGAGAGTGGAGCCTTGGTG). 

To identify mutations in *ObDMR6*, the primers Screensg442-462Fw (5′-GGTGGAGGTGGCTCATGAAT) and Screensg442-462Rv (5′-CCCTTGCTCTCCCAACACAT) flanking both sgRNA442 and sgRNA462 target sites (Figure 4) were used. 

PCR was performed using Q5 high fidelity DNA polymerase (NEB) and PCR products were cloned directly in pJET1.2/blunt vector (ThermoFisher^®^ Scientific Inc., Waltham, Massachusetts, U.S.), followed by Sanger sequencing using primers provided by the cloning kit. At least three colonies for each clone have been sequenced.

Nucleotide sequence alignment was generated using CLUSTALW tool of Bioedit 7.2 software (https://bioedit.software.informer.com, access on 15 December 2021) and the detection of insertion–deletion mutations (indels) was performed.

### 4.8. Phenotypic Evaluation of T0 and T1 Mutant Plants 

Newly regenerated transgenic T0 seedlings were in vitro multiplied and rooted. Phenotypic consequences of transformation were observed in primary transgenic plants (T0), which were potted and acclimatized in insect-proof greenhouse. Phenotypic traits, such as plant height, internode number and length, leaf morphology, length, and width, were evaluated and measured at the vegetative stage, on almost three plants of each clone. 

Selfing bags were mounted on flower stalks of T0 plants, at the beginning of flowering time, to avoid cross-pollination and mature seeds were harvested.

Fifty seeds obtained from self-pollination of T0 plants 21D and 22B were potted and grown in greenhouse in comparison to wild type plants (‘FT Italiko’) at the same age. The aforementioned phenotypic traits were also recorded and statistically evaluated, using Student’s *t*-test, in fifteen T1 plants, arising from the two clones.

### 4.9. Evaluation of In Vivo T0 Plants Infection

Preliminary infection trial with *P. belbahrii* on T0 plants to evaluate the acquired resistance through editing of the *ObDMR6* was performed under controlled conditions at CeRSAA (Albenga, Italy).

Suspension of spores of known concentration (10^6^/mL) obtained from affected fresh leaves showing evident signs of sporulation was used as inoculum and it was sprayed directly on in vivo T0 plants. Symptoms were recorded at 7-, 10- and 14-days post inoculation (dpi). Disease Incidence (DI: % of infected leaves) and Disease Severity (DS: % of infected leaf surface) were calculated. Infection (sporulation) was evaluated in Selected T0 clones (17 A2, 47 C, 21D, 45 B, and 22B) and compared with wild type (WT) ‘Italiko FT’ plant. The infection trial was performed 3 times and values of percentage are the averages of 3 replicates.

### 4.10. Volatile Profile Analysis 

#### 4.10.1. Headspace Solid Phase Microextraction Analysis

The headspace solid phase microextraction (HS-SPME) was used for the analyses of the spontaneous volatile emission of the *O. basilicum* plants. All the samples from in vitro T0 and WT plants (5 leaves each) were put in a 100 mL glass beaker covered with aluminum foil and left to equilibrate for at least 30 min. Then, the headspaces (HSs) were sampled with a Supelco PDMS fiber (100 μm) (Supelco analytical, Bellefonte, PA, USA), preconditioned according to the manufacturer’s guidelines, for 10 min at room temperature, and once the exposure time was finished the fiber was withdrawn into the needle and suddenly injected into the GC-MS equipment. The analyses were accomplished in triplicates. 

#### 4.10.2. Gas Chromatography–Mass Spectrometry Analyses 

The gas chromatography–electron impact mass spectrometry (GC–EIMS) analyses were performed with an Agilent 7890B gas chromatograph (Agilent Technologies Inc., Santa Clara, CA, USA) equipped with an Agilent HP-5MS capillary column (30 m × 0.25 mm; coating thickness 0.25 μm) and an Agilent 5977B single quadrupole mass detector using the following analytical conditions: oven temperature ramp rising from 60 to 240 °C at 3 °C/min; injector temperature at 250 °C; transfer line temperature at 240 °C; carrier gas helium, with a 1 mL/min flow. The acquisition parameters were set as follows: full scan; scan range: 30–300 m/z; scan time: 1.0 s. The identification of the constituents was based on a comparison of the retention times with those of the authentic samples, comparing their linear retention indices relative to the series C8-C27 of *n*-hydrocarbons. Moreover, computer matching was used against commercial (NIST 14 and ADAMS 2007) and laboratory-developed mass spectra libraries built up from pure substances and components of commercial essential oils of known composition and MS literature data [69]. 

#### 4.10.3. Statistical Analysis

Multivariate statistical analyses with the Principal Component Analysis (PCA) and the Hierarchical Cluster Analysis (HCA) methods were performed on the complete chemical composition of the spontaneous volatile emissions of the six *O. basilicum* samples. The PCA was carried out on a 36 × 6 covariance data matrix (36 compounds × 6 samples = 216 data), selecting the two highest PCs obtained by the linear regressions, and the chosen PC1 and PC2 covered 61.30 and 25.90% of the variance, respectively, for a total explained variance of 87.20%. The HCA was performed using Ward’s method, with squared Euclidean distances as a measure of similarity. Moreover, the statistically significant differences between content of the chemical classes in the analyzed samples were also evaluated with the analysis of variance (ANOVA) test. Averages were separated by Tukey’s post hoc test and a *p* < 0.05 was used to assess the significance of differences between means. 

The statistical analyses were performed using the JMP Pro 14.0.0 software package (SAS Institute, Cary, NC, USA).

## Figures and Tables

**Figure 1 plants-12-02395-f001:**
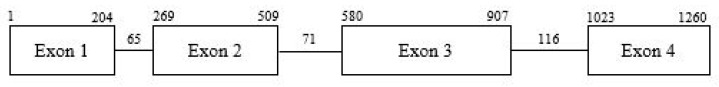
Schematic representation of *ObDMR6*: exons are visualized as boxes, and introns as lines. Numbers above the boxes indicate the starting and stop nucleotide position of cds. Length of introns is visualized above the lines.

**Figure 2 plants-12-02395-f002:**
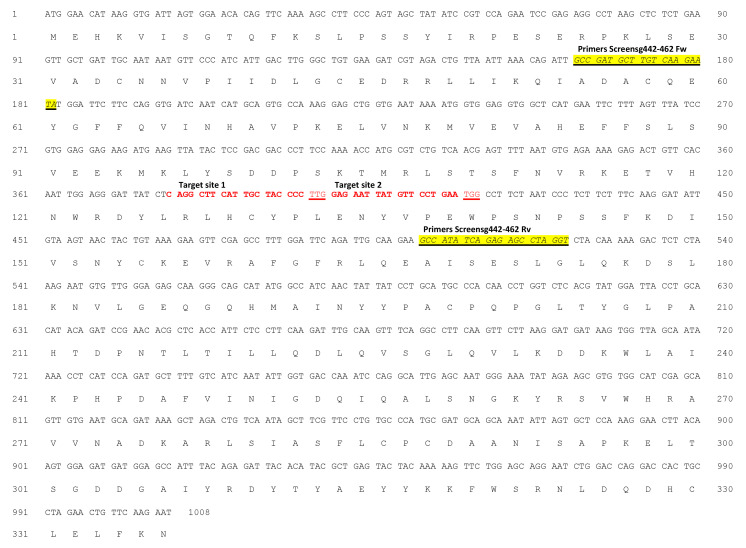
*ObDMR6* cds of sweet basil ‘FT Italiko’ and the corresponding translated amino acid sequence with the two sgRNA target sites (Target site 1: sgRNA442 and Target site 2: sgRNA442) marked in red bold and underlined PAM sites. The primers (Screensg442-462Fw and Screensg442-462Rv) used to amplify the 396 bp fragment are underlined and highlighted in yellow.

**Figure 3 plants-12-02395-f003:**
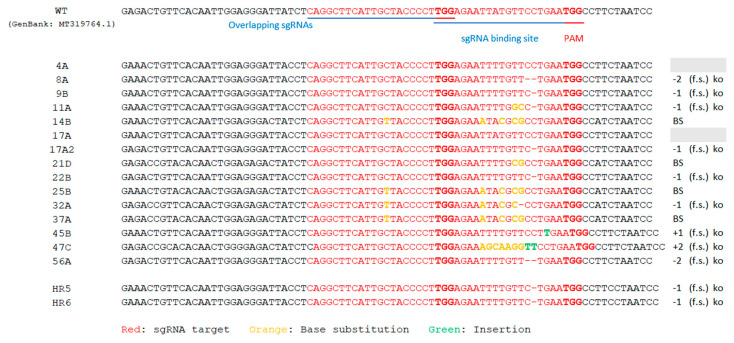
Mutation profile of individual T0 *ObDMR6*-edited sweet basil ‘FT Italiko’ plants (listed in the first column) and two hairy roots lines (HR 5 and 6). “+” or “−” indicate addition or deletion of nucleotide base. “f.s.” indicates frameshift mutations; PAM (TGG) indicates protospacer adjacent motif (in red and bold letters). The two overlapping sgRNA (sgRNA442 and sgRNA462) are underlined in blue on WT (wild type) sequence.

**Figure 4 plants-12-02395-f004:**
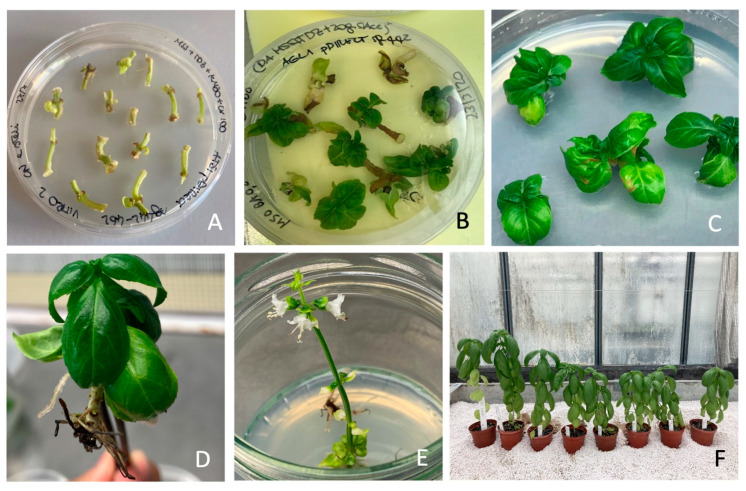
Genetic transformation and direct regeneration of *O. basilicum* ‘FT Italiko’. (**A**) Cotyledonary nodes (CNs) after co-cultivation with *A. tumefaciens* strain AGL1 in dark at 23 ± 1 °C; (**B**) CNs with regenerated shoots on regeneration medium, with cefotaxime and kanamycin; (**C**) propagation of regenerated selected clones; (**D**) well-developed and rooted in vitro edited plant; (**E**) flowering of in vitro edited plant; (**F**) acclimatized T0 basil plants.

**Figure 5 plants-12-02395-f005:**
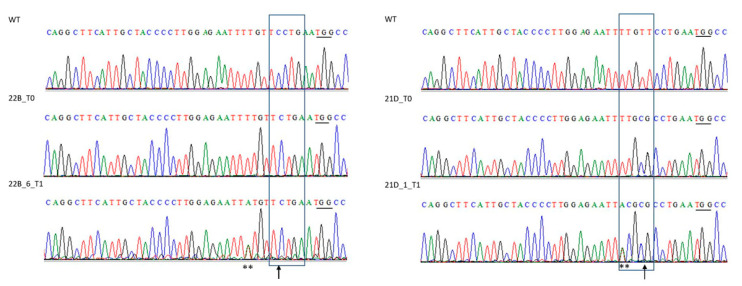
Chromatogram comparison of a 396 bp fragment of the *ObDMR6* gene from WT, T0 and plants T1 derivatives. Black boxes represent the mutation zone and in particular arrows indicate specific mutations site obtained with editing, ** indicates heterozygous A/T single nucleotide polymorphisms (SNP). Black underlines are PAM (TGG) sequence.

**Figure 6 plants-12-02395-f006:**
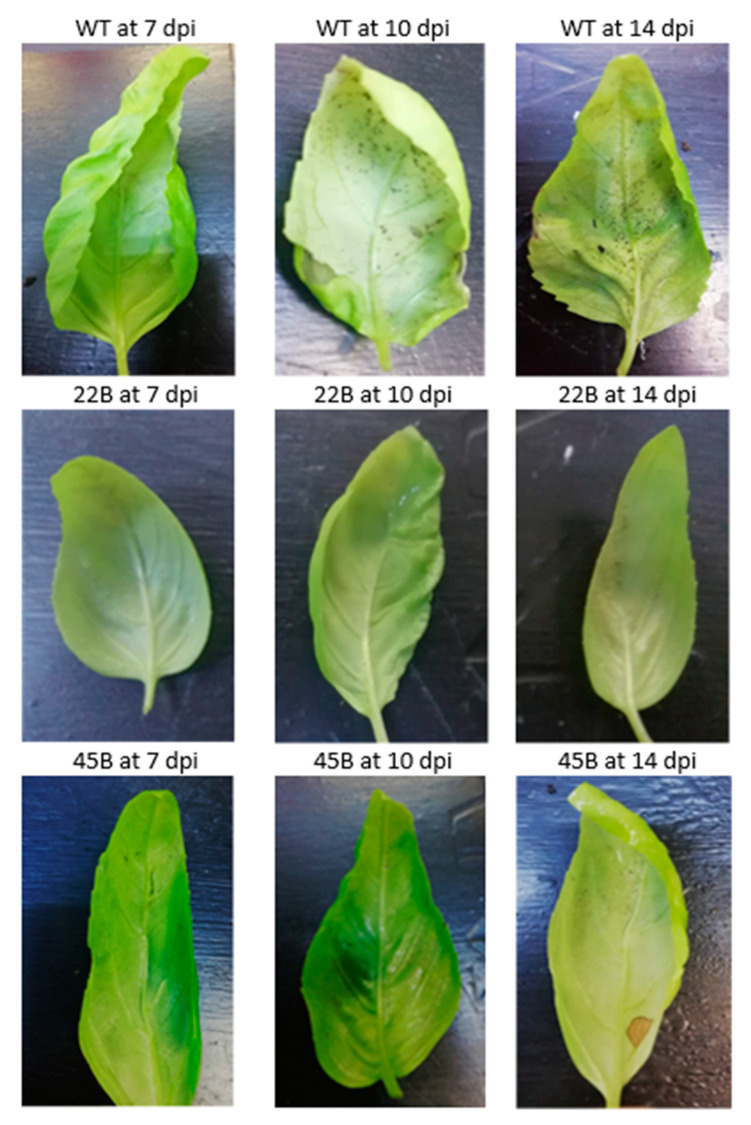
Progression of the infection on WT (control), 22B and 45 B clones at 7, 10, and 14 dpi.

**Figure 7 plants-12-02395-f007:**
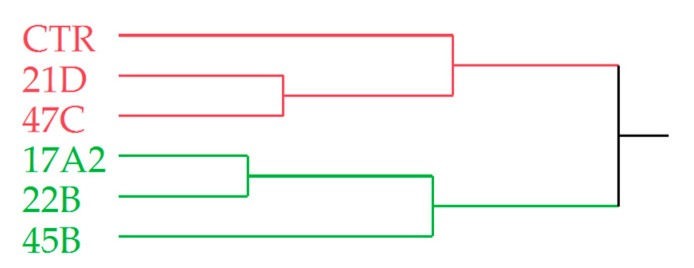
Dendrogram of the HCA performed on the complete chemical compositions of the *O. basilicum* samples.

**Figure 8 plants-12-02395-f008:**
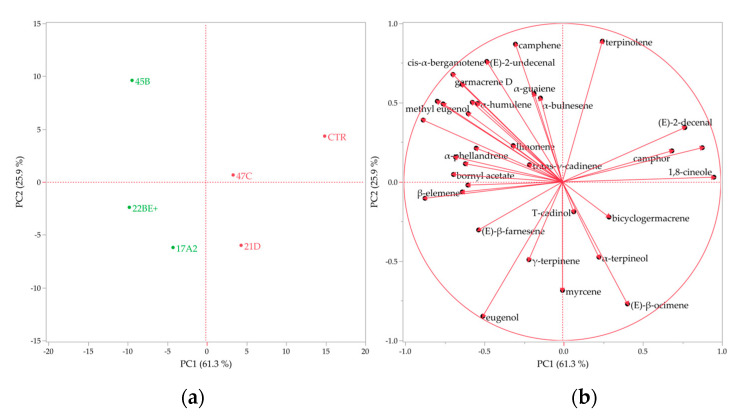
Score (**a**) and loading (**b**) plots of the PCA performed on the complete chemical compositions of the *O. basilicum* samples.

**Table 1 plants-12-02395-t001:** Morphological observations of T0 plants and description of their main traits with respect to wild type (WT).

	Leaf Traits			Plant Traits		
**Clones**	**Morphology**	**Size (cm)** **3rd True Leaf**	**Height (cm)**	**Internode** **Number**	**Internode** **Length (cm)**	**Flowering** **Attitude**
WT	Ovate with moderately serrated margin	Length: 7.75 ± 0.35 Width: 4.25 ± 0.35	48.5 ± 4.95	12.0 ± 0.41	4.04 ± 0.41	Yes
22B	Few leaves with curling up margin	Similar to WT	Similar to WT	Similar to WT	Similar to WT	Yes
21D	Few leaves with curling up margin	Shorter with decreased length/width ratio	Strongly reduced	Decreased	Shorter	Yes
45B	Similar to WT	Shorter with decreased length/width ratio	Shorter	Decreased	Similar to WT	Yes
17A2	Few leaves with curling up margin	Similar to WT	Shorter	Decreased	Similar to WT	No
47C	Few leaves with curling up margin	Shorter with decreased length/width ratio	Strongly reduced	Decreased	Shorter	No
32A	Few leaves with curling up margin	Similar to WT	Shorter	Similar to WT	Shorter	No

**Table 2 plants-12-02395-t002:** Morphological observations of T1 plants and description of their main traits with respect to wild type (WT).

	Leaf Traits		Plant Traits		
**Progenies**	**Morphology**	**Size (cm)** **3rd True Leaf**	**Height (cm)**	**Internode** **Number**	**Internode Length (cm)**
WT	Ovate with moderately serrated margin	Length: 6.4 ± 0.89 Width: 4.5 ± 0.35	27.60 ± 3.29	7.40 ± 0.55	3.73 ± 0.26
22B	Similar to WT	Higher	Similar to WT	Similar to WT	Longer
21D	Few leaves with curling up margin	Similar to WT	Similar to WT	Similar to WT	Similar to WT

**Table 3 plants-12-02395-t003:** Percentage of infected leaves (Disease Incidence) at 7-, 10- and 14-days post inoculation (dpi) (different letters indicate significant difference among clones within dpi (*p* ≤ 0.05, Tukey HSD test).

Clone	7 dpi	10 dpi	14 dpi
WT	10.0 ab	96.7 e	100.0 b
17A2	10.0 ab	93.3 e	100.0 b
47C	50.0 d	75.0 d	100.0 b
21D	23.1 c	48.7 c	100.0 b
45B	20.0 bc	23.3 b	50.0 a
22B	0.0 a	6.7 a	41.7 a

**Table 4 plants-12-02395-t004:** Percentage of infected leaves surface (Disease Severity) at 7-, 10- and 14-days post inoculation (dpi) (different letters indicate significant difference among clones within dpi (*p* ≤ 0.05, Tukey HSD test).

Clone	7 dpi	10 dpi	14 dpi
WT	28.3 bc	90.0 d	100.0 b
17A2	43.3 c	90.0 d	100.0 b
47C	30.0 bc	60.0 c	100.0 b
21D	33.3 bc	65.0 c	100.0 b
45B	20.0 bc	21.7 b	45.0 a
22B	0.0 a	10.0 a	43.3 a

**Table 5 plants-12-02395-t005:** Complete chemical composition of the samples analyzed by HS-SPME. (Different letters indicate significant differences at *p* < 0.05 (Tukey HSD test’s results).

**Compounds**	**l.r.i. ^1^**	**Class.**	**Relative Abundance ± Standard Deviation (*n* = 3)**					
			**WT/CTR**	**17A2**	**21D**	**22B**	**45B**	**47C**
α-pinene	933	mh	0.8 ± 0.01	1.5 ± 0.05	1.6 ± 0.25	1.9 ± 0.27	2.2 ± 0.21	2.3 ± 0.78
camphene	948	mh	0.5 ± 0.03	0.4 ± 0.03	0.3 ± 0.02	0.5 ± 0.05	0.7 ± 0.01	0.6 ± 0.21
sabinene	973	mh	1.3 ± 0.05	1.9 ± 0.11	2., ± 0.4	2.0 ± 0.25	2.2 ± 0.26	2.4 ± 0.55
β-pinene	977	mh	2.2 ± 0.06	3.5 ± 0.17	3.6 ± 0.72	3.9 ± 0.43	4.5 ± 0.54	4.9 ± 1.40
myrcene	991	mh	1.7 ± 0.02	2.8 ± 0.15	3.1 ± 0.25	2.7 ± 0.00	1.6 ± 0.57	3.7 ± 0.98
α-phellandrene	1006	mh	- ^2^	0.1 ± 0.02	-	-	0.1 ± 0.02	-
δ-3-carene	1011	mh	-	0.1 ± 0.01	-	-	0.2 ± 0.03	-
limonene	1029	mh	1.4 ± 0.01	1.5 ± 0.2	1.6 ± 0.43	1.7 ± 0.18	1.8 ± 0.03	2.1 ± 0.31
1,8-cineole	1031	om	44.1 ± 0.4	34.2 ± 6.02	41.2 ± 8.3	29.9 ± 1.79	33.1 ± 5.92	41.0 ± 4.21
(*E*)-β-ocimene	1047	mh	7.3 ± 0.00	7.9 ± 1.10	11.2 ± 1.33	6.3 ± 0.70	3.2 ± 1.19	4.8 ± 0.97
γ-terpinene	1058	mh	-	0.2 ± 0.06	-	-	-	-
terpinolene	1089	mh	1.5 ± 0.01	1.2 ± 0.14	1.2 ± 0.36	1.3 ± 0.05	1.5 ± 0.06	1.5 ± 0.21
linalool	1101	om	16.6 ± 0.26	2.9 ± 0.06	3.9 ± 1.47	1.8 ± 1.34	1.0 ± 0.56	4.2 ± 0.65
camphor	1145	om	3.5 ± 0.02	1.8 ± 0.22	1.7 ± 0.19	1.7 ± 0.33	1.5 ± 0.61	1.2 ± 0.54
α-terpineol	1191	om	0.3 ± 0.03	0.3 ± 0.03	0.4 ± 0.04	0.3 ± 0.09	0.3 ± 0.03	0.3 ± 0.04
(*E*)-2-decenal	1261	nt	0.1 ± 0.07	-	-	-	-	-
bornyl acetate	1286	om	-	0.2 ± 0.09	0.2 ± 0.23	0.2 ± 0.02	0.3 ± 0.32	0.3 ± 0.03
(*E*)-methyl geranate	1325	om	-	-	-	-	0.2 ± 0.06	-
eugenol	1357	pp	7.0 ± 0.02	20.0 ± 5.6	15.4 ± 7.38	18.7 ± 2.96	8.6 ± 2.41	11.2 ± 3.48
(*E*)-2-undecenal	1367	nt	-	-	-	-	0.1 ± 0.14	-
α-copaene	1376	sh	-	-	-	0.2 ± 0.04	0.3 ± 0.09	0.2 ± 0.01
β-elemene	1392	sh	0.2 ± 0.01	0.4 ± 0.07	0.3 ± 0.16	0.4 ± 0.15	0.4 ± 0.28	0.5 ± 0.28
methyl eugenol	1405	pp	3.7 ± 0.02	7.0 ± 1.28	1.8 ± 0.42	10.7 ± 5.15	15.4 ± 5.79	2.0 ± 0.45
*cis*-α-bergamotene	1416	sh	-	-	-	-	0.2 ± 0.03	-
β-caryophyllene	1419	sh	-	0.1 ± 0.02	-	0.1 ± 0.04	0.2 ± 0.10	0.2 ± 0.03
*trans*-α-bergamotene	1436	sh	5.8 ± 0.03	8.8 ± 0.69	8.5 ± 2.70	12 ± 3.92	16.5 ± 5.13	12.6 ± 2.32
α-guaiene	1439	sh	0.2 ± 0.01	0.2 ± 0.02	-	0.2 ± 0.12	0.3 ± 0.14	0.4 ± 0.09
α-humulene	1453	sh	0.2 ± 0.02	0.3 ± 0.06	0.1 ± 0.01	0.3 ± 0.07	0.4 ± 0.17	0.4 ± 0.05
(*E*)-β-farnesene	1458	sh	0.4 ± 0.05	0.6 ± 0.22	0.4 ± 0.13	1.3 ± 0.16	0.4 ± 0.16	0.6 ± 0.42
germacrene D	1481	sh	0.6 ± 0.00	0.7 ± 0.1	0.6 ± 0.33	0.9 ± 0.18	1.3 ± 0.47	0.9 ± 0.25
*trans*-β-bergamotene	1492	sh	0.2 ± 0.00	0.3 ± 0.04	0.3 ± 0.11	0.4 ± 0.12	0.5 ± 0.19	0.3 ± 0.12
bicyclogermacrene	1496	sh	0.1 ± 0.01	0.2 ± 0.01	-	-	-	0.2 ± 0.04
α-bulnesene	1505	sh	0.3 ± 0.01	0.3 ± 0.04	0.2 ± 0.11	0.3 ± 0.12	0.4 ± 0.23	0.5 ± 0.17
*trans*-γ-cadinene	1514	sh	0.2 ± 0.02	0.2 ± 0.04	0.3 ± 0.18	0.3 ± 0.12	0.3 ± 0.15	0.4 ± 0.02
β-sesquiphellandrene	1524	sh	-	0.2 ± 0.03	0.1 ± 0.08	0.2 ± 0.04	0.2 ± 0.10	0.2 ± 0.07
T-cadinol	1641	os	-	0.1 ± 0.01	-	-		0.2 ± 0.07
			**WT/CTR**	**17A2**	**21D**	**22B**	**45B**	**47C**
Monoterpene hydrocarbons (mh)			16.6 ± 0.15 ^B^	21.1 ± 2.01 ^AB^	24.6 ± 3.75 ^A^	20.3 ± 0.53 ^AB^	18.0 ± 0.77 ^B^	22.3 ± 3.46 ^AB^
Oxygenated monoterpenes (om)			64.5 ± 0.08 ^A^	39.4 ± 6.17 ^BC^	47.4 ± 6.75 ^B^	33.8 ± 3.56 ^C^	36.3 ± 4.41 ^BC^	47.1 ± 4.26 ^B^
Sesquiterpenes hydrocarbons (sh)			8.1 ± 0.05 ^B^	12.4 ± 1.30 ^AB^	10.8 ± 3.53 ^AB^	16.2 ± 4.96 ^AB^	21.4 ± 7.21 ^A^	17.2 ± 3.85 ^AB^
Oxygenated sesquiterpenes (os)			-	0.1 ± 0.01 ^AB^	-	-	-	0.2 ± 0.07 ^A^
Phenylpropanoids (pp)			10.7 ± 0.04 ^C^	27.0 ± 6.88 ^AB^	17.2 ± 6.96 ^ABC^	29.4 ± 8.1 ^A^	24.0 ± 3.38 ^ABC^	13.2 ± 3.93 ^BC^
Other non-terpene derivatives (nt)			0.1 ± 0.07 ^AB^	- ^B^	- ^B^	- ^B^	0.1 ± 0.14 ^A^	- ^B^
Total identified (%)			100.0 ± 0.01	100.0 ± 0.00	100.0 ± 0.00	99.6 ± 0.12	99.9 ± 0.06	100.0 ± 0.01

^1^ Linear retention indices on an HP5-MS capillary column. ^2^ Not detected.

## Data Availability

The data that support the findings of this study are available on request from the first author Marina Laura. The DNA sequence presented in this study is openly available in GenBank, accession n. MT319764.

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
