# Peer review of "Highly Efficient CRISPR/Cas9 Mediated Gene Editing in Ocimum basilicum ‘FT Italiko’ to Induce Resistance to Peronospora belbahrii"

_plants, 2023, doi:10.3390/plants12132395_

Round 1
Reviewer 1 Report
The manuscript described CRISPR/Cas9 mediated gene editing in Ocmum basilicum to enhance disease resistance. Though the study is intersting and may help for cultivar breeding, the manuscript is not ready to be accepted due to its serious flaws: why the different Cas9+ plants showed quite different in disease resistance, morphological traits and aromatic profile even some of they have similar mutation sites? Another error is that the reference citation format is mixed.
Author Response
Author's Reply to the Review Report (Reviewer 1)
We gratefully thank the reviewer for his valuable help and constructive criticisms. Here in detail the reply to the questions proposed. In attachment, the revised manuscript changed according to all the three reviewers.
According to Barberini et al. (2023), seedlings’ cotiledonary nodes (CNs), the best explant to induce direct regeneration in Ocimum basilicum L. ‘FT Italiko’, derived from commercial seeds (La Semiorto Sementi®) germinated in vitro. After transformation experiment, regenerants arose from different CNs and from independent regeneration event. Then, plant materials used for editing experiments own an intra-variety variability (see paragraph 4.2., rows 468-472 of the pdf revised file).
In regenerants, off target effects on the genome that were predicted by silico tool “CRISPRdirect”, (see paragraph 4.7, rows 532-533 of the pdf revised file), could be present, but in this work, we have not taken, at this stage, them into consideration, focusing first on the P. belbharii plant resistance.
Furthermore, the Agrobacterium-mediated transformation could lead to the non-predictable integration of non-T-DNA portions of the vector (backbone) into the plant genome (Oltmanns et al., 2010) and in a different way between one explant and another.
Biochemical and morphological evaluations were carried out in primary transgenic plants (T0) as reported in paragraph 4.9. (rows 565-567 of the pdf revised file) and could show differences as a consequence of the above mentioned reasons.
From the molecular point of view, some differences in disease resistance and morphological traits in Cas9+ plants are probably due to the presence of multiple members of the DMR6-like gene family, which could show varying degree of silencing.
Johnson et al. (2022) reported that basil contains a gene family of DMR6-like genes with 8 members. Six DMR6-like genes were identified by Hasley et al. (2021) in the genome of the basil cultivar Genoveser, but they were not characterized. Analysis of the genomic locations of these genes found that the eight genes could be grouped into four homeologous pairs based on common BUSCO (Benchmarking Universal Single-Copy Orthologs) genes analyzed by Gonda et al. in 2020.
Johnson et al. (2022) speculated also that the induction of nearly all the basil DMR6-like genes in compatible interaction may be the result of manipulation of their expression by P. belbahrii to reduce salicylic acid (SA) level and enhance infection; this hypothesis is supported also by Van Damme et al. (2008) who reported that the expression of the single DMR6 gene in A. thaliana was induced by both compatible and incompatible interactions with Hyaloperonospora parasitica isolates in leaves, confirming that DMR6-like genes may have different functions arising quite different responses at various levels.
Identifying the function of each DMR6-like in basil could be useful in developing basil mutants in these genes that hydroxylate SA, leading to resitance to downy mildew (DM). Similar observations were reported by Kieu et al. (2021) in potato: two DMR6 genes were individually mutated using CRISPR-Cas9, but only the mutation in StDMR6–1 resulted in plants with increased resistance to late blight. Moreover, Zeilmaker et al. showed that AtDMR6 and AtDLO1 (DMR6-Like Oxygenases) are partly redundant in suppressing DM resistance and are involved in SA inactivation: only the dmr6-dlo1 double mutant is completely resistant to H. parasitica in Arabidopsidis, and resistance is accompanied by enhanced expression of defense-related genes, hyper-elevated SA levels, and a corresponding dwarf phenotype.
In our work, the comparison between the sequence of ObDMR6 from ‘FT Italiko’ and the O. basilicum genome sequences recently released (Gonda et al., 2020) highlighted the presence of six variable copies of the DMR6 gene in basil, being the ‘FT Italiko’ sequence used in the present study closer to copies present in genomic scaffolds 7503 and 262 (see lines 123-125 in the manuscript). Even if the two overlapping sgRNA target sequences were designed on the second exon of the gene ObDMR6 ‘FT Italiko’ and they were found to be conserved in all 6 chromosome scaffolds identified by CoGeBlast tool (see lines 358-361 in the manuscript) additional orthologues in the genome may not yet be identified and therefore they could give a different response to DM, showing a different behavior in our selected clones.
Regarding the aromatic profile, the phenylpropanoid volatiles biosynthesis pathway in tetraploid sweet basil is partially known (Gonda et al. 2020). Although in literature we haven’t found direct references related to the influence of gene editing of the DMR6 family on the biosynthesis of phenylpropanoids, we could suppose that DMR6, being able to influence the biosynthesis of salicylic acid, could in turn be able to modify the production of phenylpropanoids, with which it shares the same biosynthetic pathway (shikimate pathway).
In order to better explain all these considerations, we added the following sentences in the revised manuscript in paragraph 3 (discussion) from line 433 of of the pdf revised file:
Quite differences observed among Cas9+ plants in disease resistance, morphological traits and aromatic profile are probably due to intra-variety variability of commercial seeds as starting material for obtaining CNs and to independent origin of each regenerant plant from CNs following genetic transformation. Furthermore, other factors should be considered, as non-predictable integration of non-T-DNA portions of the vector into the plant genome (Oltmanns et al., 2010) due incorrect recognition of borders by Agrobacterium and off-target effects on the genome that were predicted only in-silico. In addition, multiple members of the DMR6-like gene family, recently identified in sweet basil (Hasley et al. 2021; Johnson et al., 2022), may contribute to show varying degree of silencing and restore partially the susceptibility to DM.
The new references added in the text and in the ref. list:
[57] Oltmanns, H.; Frame, B.; Lee, L.Y.; Johnson, S.; Li, B.; Wang, K.; Gelvin, S.B. Generation of backbone-free, low transgene copy plants by launching T-DNA from the Agrobacterium chromosome. Plant Physiol. 2010 152(3), 1158-66. https://doi.org/10.1104/pp.109.148585.
[58] Johnson, E.T.; Kim, H.S.; Tian, M.; Dudai, N.; Ta, O.l, Gonda, I. Dual transcriptional analysis of Ocimum basilicum and Peronospora belbahrii in susceptible interactions. Plant Gene 2022, 29, 100350ISSN 2352-4073, https://doi.org/10.1016/j.plgene.2021.100350.
All the references are now updated in the manuscript as requested. The final list has been carefully checked.
Reviewer 2 Report
This manuscript describes the application of CRISPR/Cas9 editing in sweet basil to induce plant resistance toward pathogens, in particular vs. Peronospera belbahrii. The authors perform DMR6 editing approaches in an élite cultivar of sweet basil and obtained several mutant lines. Leaf assay results revealed that To mutant plants showed a moderate resistance to the fungal pathogen. This research was done and described well in the manuscript. Although authors present interesting data, the only and big drawback of the manuscript is basically the absence of any real novelty. This work cannot be considered entirely new both for the idea/applied methodology and the informations derived from it. As reported by the authors themselves, Hasley et al. (2021) – Plos One 16(6) carried out a similar work with similar main results using the same plant species (Ocimun basilicum), genome editing approach (CRISPR/Cas9), target gene (ObDMR6) and main objective (pathogen resistance). Although in the manuscript of Laura et al., the authors performed for the first time a phenotypic and biochemical characterization of the DMR6 mutants, the two works differ substantially for the tested basil cultivar.
Author Response
Author's Reply to the Review Report (Reviewer 2)
We thank the reviewer for his valuable help. Here in detail the reply to the observations. In attachment, the revised manuscript changed according to all the reviewers.
The choice of varietal selection was conditioned by the simultaneous fulfillment of two conditions: (1) the conformity of the varietal selection with the morphological characteristics established by UPOV and specifically envisaged for “Genovese” basil grown in Liguria (AAVV, 2000); (2) the compliance of the varietal selection with the list of basil varieties, attributable to the "Genovese" type, which can be used for the production of Genovese Basil PDO (MADE IN QUALITY - MREC_CERTIF_65_OdC; issue date 04.16.2019 - Rev. 1).
The FT ITALIKO varietal selection of the Semiorto company satisfies both requirements. The "Genoveser" selection of the Enza Zaden company only meets the first of the two requirements. Since the work is also intended to improve the qualitative knowledge of Genoese type basil, which can also be used for the Denomination of Origin product, this choice was made.

Reviewer 3 Report
The reviewed manuscript describes the development of induced Peronospora resistance in basil by gene editing.
The topic of this study is very timely, as the extent of damage caused by pathogens has multiplied due to climate change, making the generation of new resistant genotypes desirable.
The manuscript is well structured and the proportion of chapters is appropriate. The results are well presented and discussed in detail.
In the following I would like to draw the authors' attention to some minor, mainly editorial, errors:
According to the journal's rules, only the numbers should be used in references, which should be corrected throughout the manuscript.
In line 90, the name of the genus (Arabidopsis) should be written out at the first mention of A. thaliana.
On line 99: The scientific name of each plant or the English name should be given. For potatoes, apples, and citrus, I also recommend using the Latin name (Solanum tuberosum, Malus domestica, Citrus ssp.).
Lines 105-109 (the last two sentences of the introduction) should be deleted or reworded as they already contain results.
As mentioned above, in line 151, the name of the genus (Agrobacterium) should be written out at the first mention of A. rhizogenes.
Author Response
Author's Reply to the Review Report (Reviewer 3)
We gratefully thank the reviewer for his valuable help and his support for editorial errors. Here in detail the reply to the questions proposed point by point. In attachment, the revised manuscript changed according to all the three reviewers.
- According to the journal's rules, only the numbers should be used in references, which should be corrected throughout the manuscript.
The references are now updated in the manuscript as requested. The final list has been carefully checked.
- In line 90, the name of the genus (Arabidopsis) should be written out at the first mention of A. thaliana.
We changed it in the text.
- On line 99: The scientific name of each plant or the English name should be given. For potatoes, apples, and citrus, I also recommend using the Latin name (Solanum tuberosum, Malus domestica, Citrus ssp.).
We changed it in the text.
- Lines 105-109 (the last two sentences of the introduction) should be deleted or reworded as they already contain results.
We agree to shorten this part, removing sentences referring to results and focusing only on the aim of the research.
- As mentioned above, in line 151, the name of the genus (Agrobacterium) should be written out at the first mention of rhizogenes.
We changed it in the text.

Round 2
Reviewer 1 Report
Though the phenotype of CRISPR/Cas9 editing plants are not well uniformity among different lines, the results of this manuscript are sitll interesting and worth to be published.